# Geographical Variability in *CYP1B1* Mutations in Primary Congenital Glaucoma

**DOI:** 10.3390/jcm11072048

**Published:** 2022-04-06

**Authors:** Manali Shah, Rachida Bouhenni, Imaan Benmerzouga

**Affiliations:** 1Department of Biomedical Sciences, West Virginia School of Osteopathic Medicine, Lewisburg, WV 24901, USA; mshah1@osteo.wvsom.edu; 2Department of Pharmaceutical Sciences, Northeast Ohio Medical University, Rootstown, OH 44272, USA; rabdou6@gmail.com; 3Vision Center, Akron Children’s Hospital, Akron, OH 44308, USA

**Keywords:** primary congenital glaucoma, cytochrome p450 1B1 (*CYP1B1*), mutations, demographics

## Abstract

Primary congenital glaucoma (PCG) is a rare type of glaucoma that is inherited in an autosomal recessive manner. PCG can lead to blindness if not detected early in children aged 3 or younger. PCG varies in presentation among various populations, where disease presentation and disease severity vary by mutation. The most common gene implicated in PCG is cytochrome p450 1B1 *(CYP1B1).* Here, we sought to review the literature for mutations in *CYP1B1* and their presentation among different populations. Areas of interest include recent findings on disease presentation and potential implications on our understanding of PCG pathophysiology.

## 1. Introduction

Glaucoma is a set of ocular diseases that can affect the optic nerve and can lead to irreversible blindness. Primary congenital glaucoma (PCG) is a rare form of glaucoma and is characterized by ocular anomalies that cause defects in the aqueous humor drainage system [1]. PCG affects children age 3 or younger [2,3]. The worldwide incidence of PCG is estimated to be 1 in 10,000 to 1 in 70,000 live births, making early diagnosis a challenge in some populations [4,5]. Despite its rare nature, it accounts for 18% of childhood blindness [2]. PCG incidence varies by population, whereas disease severity varies by mutation [1,4,5]. PCG is often caused by autosomal recessive mutations in the cytochrome P450, subfamily 1, polypeptide 1 (*CYP1B1*) [1,2,6,7,8]. *CYP1B1* encodes a 543-amino-acid dioxin-inducible member of the cytochrome p450 gene superfamily [9]. The enzyme is involved in the metabolism of a variety of substrates, including steroids and retinoids that can act as morphogens during development [10]. *CYP1B1* is expressed in various human ocular tissues including the cornea, ciliary body, iris, retina [11,12] and the trabecular meshwork [13]. Other causative genes identified to date include latent transforming growth factors beta-binding protein 2 (LTBP2) and the angiopoietin receptor tunica interna endothelial cell kinase (TEK) [14,15].

In this article, we review the presentation of *CYP1B1* (the most causative gene)-related PCG in different geographical areas (countries) and the potential implications of these variabilities on the different populations. Areas of interest include (1) findings of mutations in *CYP1B1* associated with PCG, (2) variability based on geographical region and (3) implications on our understanding of the pathophysiology of PCG.

## 2. Materials and Methods

PubMed was searched from January 2012 to July 2021 for articles that included population-based studies/cross-sectional studies. The search terms were grouped in 16 combinations (Appendix A). The first search on “PCG and mutation” found 231 articles and dated back to 1984. This search was further filtered to include articles since 2012 (articles published in the past 10 years), resulting in a total of 117 articles. To be more specific and to answer the question about *CYP1B1* mutation and its demographics, the following specific keywords were included: *CYP1B1*, ethnicity, developmental, consanguinity, demographic, etc. The search included patients from prenatal to 3 years of age. Articles about adult glaucoma, open and closed angle glaucoma, pharmacology, pathobiology, etc. were excluded (Figure 1).

## 3. Results

### 3.1. Primary Congenital Glaucoma and Clinical Presentation

Primary congenital glaucoma is an ocular disorder inherited in an autosomal recessive manner, where *CYP1B1* was the first gene to be described as the causative agent for PCG [6,13] and the most commonly mutated gene in PCG [16]. The prevalence of the disease varies based on population, where the incidence is as high as 1 in 1250 in a subpopulation of Slovakia [17] or as low as 1 in 20,000 in Western countries [3]. Patients with mutations in *CYP1B1* exhibit variable disease presentation, even within the same population and the same mutation [1]. The protein belongs to the cytochrome p450 enzyme family and is involved in the metabolism of a variety of substrates, including steroids and retinoids that can act as morphogens during development [10]. The most accepted theory regarding the pathophysiology of PCG states that mutations in *CYP1B1* lead to a halt in the migration of the embryonic neural crest cells destined to become trabecular meshwork cells during development, leading to a maldeveloped and dysfunctional trabecular meshwork [17], thus causing the buildup of aqueous humor fluid within the limited space of the anterior chamber. This raises the intraocular pressure, compressing the optic nerve, resulting in blindness if not corrected surgically (Figure 2). Congenital glaucoma registry at King Khaled Eye Specialist Hospital lists the prevalence rate of PCG in various regions of Saudi Arabia’s southern province (27.8%), western province (23.6%), central province (22.2%), eastern province (11.1%) and northern province (9%). The incidence of PCG in Western countries (Ireland, Britain and USA) is 1 per 10–20,000 live births [3]. Recently, a genotype–phenotype study in a cohort of PCG patients in Morocco showed that disease severity, including mean intraocular pressure and number of surgeries, was significantly higher in the *CYP1B1* mutation carriers as well as the double *CYP1B1* null alleles [18]. Therefore, mutations in *CYP1B1* are often associated with the most severe disease compared to other genes that are associated with the development of PCG [19].

### 3.2. CYP1B1 and PCG in Different Countries

Recently, mutations in *CYP1B1* were shown to have an 80–100% prevalence in Saudi Arabians and Slovakian Rom populations, respectively [25,26,27,28]. The point mutation p. Glu387Lys (E387K) accounts for all pathogenic variants in the Slovakian Rom population [26], and p. Gly61Glu (G61E) accounts for most variants in Saudi Arabians [27,28]. A younger age at onset and positive family history of PCG was associated with any mutation in *CYP1B1* in a large Indian cohort sample [29]. The Chinese Han population holds a lower incidence (17.2%) of PCG caused by mutations in *CYP1B**1* [30]. Two mutations in *CYP1B1* in the Han Chinese, p. Ala330Phe (A330F) and p. Arg390His (R390H), alter the protein structure of *CYP1B1,* resulting in a severe form of PCG where even multiple surgeries and medications are not effective [31]. In the US, five novel heterozygous variants were identified in seven families with PCG that were clinically severe. None of the variants were the most common mutation in other populations [32]. In Nouakchott, a white Maure family blinded by glaucoma carried a homozygous mutation in *CYP1B1* that was not present in other families and results in an early terminating codon at position 150 downstream of *CYP1B1,* resulting in a heavily shortened *CYP1B1* [33]. The prevalence of this mutation in *CYP1B1* among the white Maure ethnic group remains unclear. Table 1 summarizes the point mutation findings in *CYP1B1* most commonly found in the reviewed papers and the prevalence of *CYP1B1* mutation based on population.

### 3.3. Gender and Laterality in PCG

The role of biological sex is currently unclear. Recently, in the regions of Asia, Europe, America and Africa, the role of biological sex was not significant in the presentation of PCG [33]. However, a study performed in western Saudi Arabia found more males (61.8%) were affected with PCG than females (38.2%) [27]. The latter finding is further supported by a different study in Istanbul, Turkey, as they also found males (57.2%) to be more affected than females (42.8%) [34]. Most of the researchers studied the bilaterality of the disease and found that the probability of identifying biallelic pathogenic variants in PCG caused by *CYP1B1* increased with the presence of severe disease, a positive family history for the disease and parental consanguinity [3]. A recent study evaluated the risk factors associated with the development of PCG and found a significant association to material health, race/ethnicity and low-term birth weight [35]. Collectively, this further supports the idea that the disease severity varies by populations/geographic areas where the impact of these factors is of significance.

**Table 1 jcm-11-02048-t001:** Common mutation/s for *CYP1B1* pathogenic variants with their respective country or ethnicity. Prevalence of mutations in *CYP1B1* as reported in [25]. The most common mutations reported in the studies reviewed. Over 140 alterations including deletions, insertions and missense mutations identified in *CYP1B1* [36].

Country	Prevalence of Mutations in *CYP1B1* [25]	Most Common Mutation in *CYP1B1*
Pakistan	Not reported	p. Arg390His
Indonesia	33.3%	No specific variant
Japan	20%	p. Arg444Gln
Saudi Arabia	80–100%	p. Gly61Glu
		p. Glu229Lys
		p. Arg390His
South Korea	Not reported	p.Gly329Ser
India	44%	p. Ser476Pro
Brazil	50%	p. Arg368His
		p. Gly61Glu
Morocco	Not reported	p. Gly61Glu
Iran	70%	p. Arg469Trp
		p. Arg368His
		p. Arg390His
		p.Gly61Glu
		p. Glu173Arg
India, China, Iran	Not reported	p. Arg390His
Slovakian Roma	80–100%	p. Glu387Lys
Hungarian	Not reported	
Portugal, Vietnam	Not reported	p. Gly61Glu
Chinese Han	Not reported	p. Arg330Phe and p. Arg390His

## 4. Discussion

In this review paper, the most common *CYP1B1* mutations identified in various populations are summarized in Table 1. The early diagnosis of PCG can prevent blindness and irreversible damage to the ocular structures. It is known that *CYP1B1* plays a role in the metabolism of many compounds such as estrogen, melatonin, retinoids, vitamins, fatty acids, etc. [1]. Mutations in *CYP1B1* consequently affect ocular tissues by a mechanism that remains unclear.

The studies reviewed in this article highlight the importance of wild-type *CYP1B1* in the developing eye. Specifically, mutations in *CYP1B1* were shown to be the most common in disease presentation irrespective of the population. This is consistent with previous reports that described *CYP1B1* as the most causative gene of PCG [37]. However, the type of mutation in *CYP1B1* varied by population, suggesting additional factors that contribute to the type of mutation and hence disease presentation. The most common mutation in *CYP1B1* in several populations is the p. Gly61Glu (G61E) point mutation; however, the PCG caused by the p. Gly61Glu mutation, which results in a 50% reduced active protein [38], can be corrected surgically in these patients if identified early. Interestingly, another rare compound mutation that is found in the Chinese Han population significantly alters CYP1B1 protein structure, including changes in the ligand-binding pocket, and the resulting elevated intraocular pressure cannot be corrected medicinally or surgically. This suggests that in addition to *CYP1B1* being developmentally important to ocular tissues, the population/mutation interplay contributes to the severity of the disease phenotype. It remains unclear as to which factors may contribute to the presentation of specific mutations in *CYP1B1* compared to other mutations in specific populations. It is possible that several factors contribute to mutation/population disease presentation such as maternal health and race/ethnicity. Therefore, it is likely that these factors may influence *CYP1B1* protein expression, an area that varies between species [39], or its enzymatic activity that is necessary for the development of ocular structures [20]. *CYP1B1* is not required for mammalian development [40], and it is likely that altered enzymatic activity contributes to disease severity and that in turn can vary by population, where additional factors can contribute to the persistence of mutations of increased severity. It is worth noting that the majority of *CYP1B1* mutations reported in the literature and in our study fall within the conserved core structures of *CYP1B1* [1], thus allowing these mutations to cause a spectrum of presentations for PCG severity based on the degree of the alteration. The development of the eye is a complex process involving a number of gene products that may interact with *CYP1B1* that then lead to the variable disease presentation observed in patients with PCG [37]. In addition, a PCG phenotype that is resistant or refractory to treatment may benefit from a global understanding of patient genetics in response to current interventions.

### 4.1. Future Implication—Prescreening and Prevention

A potential preventative approach could be pre-screening measures when available. Hospitals could offer genetic screening to PCG patients or those with increased risk of PCG. Those who test positive for mutations in *CYP1B1* could either be alerted or made aware of any surgeries that might need to take place shortly after birth. Prenatal testing can be offered, but its purpose is tied with several debates, including pregnancy termination [25]. The two known factors that can place an infant at an increased risk of developing PCG are family history and consanguineous marriage. There are more familial (87%) cases being reported compared to sporadic (27%) cases [3]. Therefore, genetic screening may also provide an early diagnosis, justifying the importance of genetic/prenatal screening as a preventative measure. Prenatal diagnosis for pregnancies at increased risk is possible if the PCG-causing pathogenic variant(s) in a family are known [25].

### 4.2. Limitations

The studies included in this review were retrospective case–control studies. Future studies involving real-time or prospective data could prove to be especially beneficial. Additionally, certain phenotypes can have similar clinical presentations and therefore could be mistaken for PCG when recruiting patients into studies. Studies in the literature conducted for PCG are usually retrospective, have small sample sizes, are nonrandomized and are mostly in middle eastern regions [3]. Therefore, future studies should aim at a better understanding of the variability in disease presentation. Additionally, the timeframe selected for this search may have limited our access to large pools of specific isolated genetic data. Hence, the chosen methodology is not inclusive of all end points associated with the disease.

## 5. Conclusions

Primary congenital glaucoma is a rare developmental glaucoma that manifest during infancy and up to 3 years of age. PCG is mainly inherited in an autosomal recessive manner, thus causing an increased prevalence of PCG in populations/countries with higher likelihoods of inbreeding. Among Asians, Saudi Arabians, and Indians, the average that a child becomes symptomatic is around 3–4 months, whereas in Western countries, it is 11 months. This variation is attributed to the type of mutation in *CYP1B1* and therefore the disease. However, the Chinese Han population carry the most severe compound mutations but are also among the populations with lower PCG prevalence. Collectively, these findings highlight the heterogenicity of disease presentation based on country of origin, suggesting a molecular cause for PCG that can be influenced by patient group or population.

## Figures and Tables

**Figure 1 jcm-11-02048-f001:**
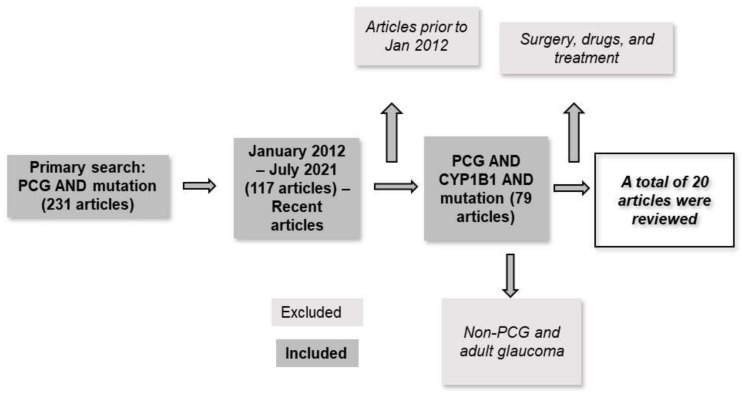
Flow chart of the search strategy steps for the reviewed articles. PubMed was searched from 2012 to 2021. A total of 20 articles were reviewed.

**Figure 2 jcm-11-02048-f002:**
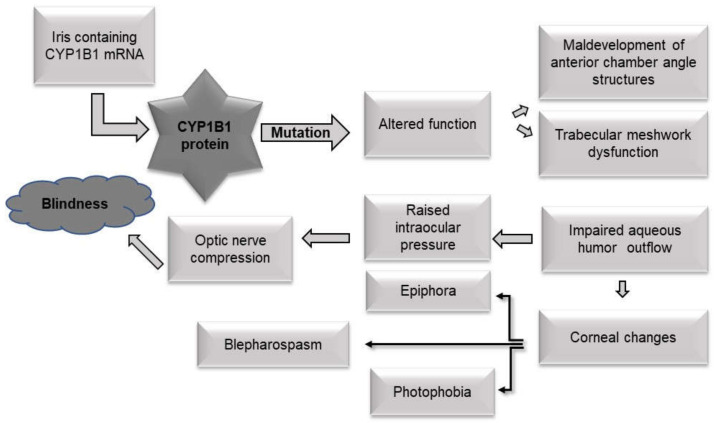
The pathophysiology of primary congenital glaucoma (PCG). The most accepted theory for the pathophysiology of PCG resulting from the failure of the CYP1B1 downstream pathways in the development of functional ocular structures. Early blindness results from damage to the optic nerve due to elevated intraocular pressure (IOP) [20,21,22,23,24].

## Data Availability

Not applicable.

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
