# Peer review of "Geographical Variability in *CYP1B1* Mutations in Primary Congenital Glaucoma"

_jcm, 2022, doi:10.3390/jcm11072048_

Round 1
Reviewer 1 Report
Dear Authors,
This is a well-organized review with a crucial focus the presentation of CYP1B1 (the most causative gene) related PCG in different geographical areas. Given the population heterogeneity of PCG and the increasing number of mutations found in CYP1B1, it is of great importance to summarize and analyze common CYP1B1 mutations for disease prevention and diagnosis.
There are no major problems with this review; the data are very well presented in text, tables and graphs.
Here are my comments and suggestions.
- Section 3.1: There is lack of information about the loci of CYP1B1 gene and structure of CYP1B1 An explanation of the role of Cytochrome P450B1 in eye development should be added.
- In Section 3.2: There is information about A330F mutation found in Han Chinese population in text. In Table 1 it is not shown. Why?
- In Section 3.2: Table 1 shows only missense mutations related with PCG. However, deletions, insertions or duplication in CYP1B1 gene are also common.
- Disscussion: Discussion of the possible effects of the mutation found in the Chinese Han population (the most severe) and the mutation (Gly61Glu) found in the remaining populations on the structure and function of cytochrome P4501B1 is needed. This is important because one of the goals of this article is to understand the pathophysiology of PCG.
- Keep the three-letter symbols for amino acids in mutation names in all text.
- In general, symbols for genes are italicized. Consistency is needed in the presentation of gene names in the text.
Author Response
Response to Reviewer 1:
This is a well-organized review with a crucial focus the presentation of CYP1B1 (the most causative gene) related PCG in different geographical areas. Given the population heterogeneity of PCG and the increasing number of mutations found in CYP1B1, it is of great importance to summarize and analyze common CYP1B1 mutations for disease prevention and diagnosis.
There are no major problems with this review; the data are very well presented in text, tables and graphs.
Here are my comments and suggestions.
- Section 3.1: There is lack of information about the loci of CYP1B1 gene and structure of CYP1B1 An explanation of the role of Cytochrome P450B1 in eye development should be added.
This is an excellent suggestion. We added elaboration about CYP1B1 into the introduction section (section 1- highlighted). We also added elaboration to section 3.1 as you suggested in line 57-65 (highlighted).
- In Section 3.2: There is information about A330F mutation found in Han Chinese population in text. In Table 1 it is not shown. Why?
Good question. We have added it to the table, the reason it was not added is the prevalence information comes from a different paper, not the one referenced for the prevalence column in Table 1. However, we have included it without the prevalence and the prevalence is in the text citing the paper with that information.
- In Section 3.2: Table 1 shows only missense mutations related with PCG. However, deletions, insertions or duplication in CYP1B1 gene are also common.
This is a great comment, thank you. We have added a reference that has summarized the various alterations in CYP1B1 (Haddad, et.al, 2021) in the table legend for more CYP1B1 mutations (line127). We also added a sentence to the legend to indicate there are over 140 mutations identified in CYP1B1 to date, this hopefully will clarify our table is a brief summary, which then allows us to highlight variability across population.
Discussion: Discussion of the possible effects of the mutation found in the Chinese Han population (the most severe) and the mutation (Gly61Glu) found in the remaining populations on the structure and function of cytochrome P4501B1 is needed. This is important because one of the goals of this article is to understand the pathophysiology of PCG.
Thank you for this very helpful suggestion. We have added line 152-165 (highlighted) for additional substance for discussion.
- Keep the three-letter symbols for amino acids in mutation names in all text.
Thank you, that has been completed.
- In general, symbols for genes are italicized. Consistency is needed in the presentation of gene names in the text.
Thank you. We have checked that all genes are italicized.

Reviewer 2 Report
The authors report a review of the literature on CYP1B1 variants associated with primary congenital glaucoma (PCG) and the variability among different populations. It is unclear to me why the authors conducted a search from January 2012 when CYP1B1 variants have been identified in association with PCG in 1998 ad many publications on the variability of CYP1B1 variants in different populations has been published between 1998 and 2012. Moreover, it is unclear to me how the search led to so few articles when there is a wealth of publications on CYP1B1 and PCG across several additional populations than presented here, even since 2012. A quick search of the literature led to many more papers that should have been included (for example PMIDs 24940937, 33745036, 30108387, 24942078, 25952714, 30520782, 26550974 - and this is by no mean an extensive list of papers not included).
The manuscript is quite unthorough. For example, the authors list a worldwide incidence of PCG of 1/10,000 to 1/70,000 but incidences below 1/10,000 have been reported in several populations, including as low as 1/1,250. They further discuss additional incidence of the disease under the clinical presentation section of their results which seems misplaced. The authors should cite original research instead of review papers (e.g. no reference is provided for the TEK gene and its association with PCG, GeneReviews is cited instead of original papers, a single review paper is cited for the fact that CYP1B1 variants often cause a more severe disease instead of the original papers).
Half of the mechanisms presented in Figure 2 are not discussed in the paper and no references are cited to support them (e.g. failure of steroid metabolism, maldevelopment of the anterior chamber angle structures, corneal changes).
The incomplete review of the literature is also reflected in the discussion of CYP1B1 variants among different populations. For example, although initial studies reported a high rate of CYP1B1 variants in Rom Slovakian populations, further studies have showed a higher variability and genetic heterogeneity in other Gypsy populations, which is not discussed here. Similarly, p.G61E does not account for all variants in Saudi Arabians, further studies have showed that there are other variants than p.G61E. Table 1 should be listing all populations with CYP1B1 variants and a more comprehensive list of variants identified in each population than just the few listed here since it is the topic of the manuscript.
The discussion lacks a more global review of the gene and its variants and instead provides specific examples of variants/populations to make more general claims. For example, the fact that patients with a specific variant in a Chinese population were more refractory to treatment does not necessarily mean that variants contribute to severity. A more thorough review of the literature across populations, variant type and surgical procedures and outcomes would be needed to address the potential severity and variability of CYP1B1 variants, which was not presented here. The abstract mentions that this is a review of CYP1B1 variants and their presentation among different populations. However, the discussion fails to address this in a comprehensive manner.
The HGVS recommendations (http://varnomen.hgvs.org/) should be followed for gene and variant nomenclature. The new terminology recommends using terms such as “variant” or “alteration” instead of “mutation”. Gene names should be in italic.
Author Response
Response to Reviewer 2:
The authors report a review of the literature on CYP1B1 variants associated with primary congenital glaucoma (PCG) and the variability among different populations. It is unclear to me why the authors conducted a search from January 2012 when CYP1B1 variants have been identified in association with PCG in 1998 ad many publications on the variability of CYP1B1 variants in different populations has been published between 1998 and 2012. Moreover, it is unclear to me how the search led to so few articles when there is a wealth of publications on CYP1B1 and PCG across several additional populations than presented here, even since 2012. A quick search of the literature led to many more papers that should have been included (for example PMIDs 24940937, 33745036, 30108387, 24942078, 25952714, 30520782, 26550974 - and this is by no mean an extensive list of papers not included).
Thank you for the comments and valid points. To address your comments regarding the methodology, some text was added to the material and methods (line 47) to indicate that our range was selected to contain most recent reports to identifyany newer findings, however, our search results largely agree with the findings in the original literature. We also added to section 3.1 the references where CYP1B1 was identified as a causative gene of PCG (Safarazi, 1995). We have also increased the information about CYP1B1 in the introduction (line 27-34) as well as in the results section (line 58-70). These include some of the papers you referenced. We have also added information to the legend of the table to describe that our table is a brief summary, where there are over 140 mutations in CYP1B1 identified to date (Line 125-127), as well as the reference that conducted a meta-analysis for additional mutations. We have added additional references to our paper to encompass some of the papers you referenced as well as address your comments. Thank you very much for helping us improve the paper.
The manuscript is quite unthorough. For example, the authors list a worldwide incidence of PCG of 1/10,000 to 1/70,000 but incidences below 1/10,000 have been reported in several populations, including as low as 1/1,250. They further discuss additional incidence of the disease under the clinical presentation section of their results which seems misplaced. The authors should cite original research instead of review papers (e.g. no reference is provided for the TEK gene and its association with PCG, GeneReviews is cited instead of original papers, a single review paper is cited for the fact that CYP1B1 variants often cause a more severe disease instead of the original papers).
Thank you for the comment. We have added details about the variability of incidence in various populations in our result section (line 60+61). The prevalence data was obtained from PubMed GeneReviews, as it cites all the original papers, but original papers were cited as well. We have added the reference for the TEK and PCG (line 34), it was an oversight on our end, so thank you for thoroughly reviewing our manuscript.
Half of the mechanisms presented in Figure 2 are not discussed in the paper and no references are cited to support them (e.g. failure of steroid metabolism, maldevelopment of the anterior chamber angle structures, corneal changes).
Thank you for this comment. We have adjusted the failure of steroid metabolism to altered function for an appropriate representation of the current literature. We also added the references to the body of the legend for the figure (line 88). We also referenced the original paper with the hypothesis in the text of the figure (line 68)
The incomplete review of the literature is also reflected in the discussion of CYP1B1 variants among different populations. For example, although initial studies reported a high rate of CYP1B1 variants in Rom Slovakian populations, further studies have showed a higher variability and genetic heterogeneity in other Gypsy populations, which is not discussed here. Similarly, p.G61E does not account for all variants in Saudi Arabians, further studies have showed that there are other variants than p.G61E. Table 1 should be listing all populations with CYP1B1 variants and a more comprehensive list of variants identified in each population than just the few listed here since it is the topic of the manuscript.
Thank you for the comments. We agree and there are numerous papers that show other mutations in these populations. We are not claiming that this is the only mutation found, rather the most common reported in the literature. We have changed the “all” to most to address this miscommunication on our end for the Saudi Population and added the reference. We also added the reference for the Slovakian Rom. In addition, text was added to address the list of mutations known (over 140), as well as the reference that conducted a thorough summary of all variants across different populations (Haddad, et. al, 2021). We added text to the discussion to address the contributing factors to cross- population variability (line 152-165).
The discussion lacks a more global review of the gene and its variants and instead provides specific examples of variants/populations to make more general claims. For example, the fact that patients with a specific variant in a Chinese population were more refractory to treatment does not necessarily mean that variants contribute to severity. A more thorough review of the literature across populations, variant type and surgical procedures and outcomes would be needed to address the potential severity and variability of CYP1B1 variants, which was not presented here. The abstract mentions that this is a review of CYP1B1 variants and their presentation among different populations. However, the discussion fails to address this in a comprehensive manner.
The HGVS recommendations (http://varnomen.hgvs.org/) should be followed for gene and variant nomenclature. The new terminology recommends using terms such as “variant” or “alteration” instead of “mutation”. Gene names should be in italic.
Thank you for the comments. You raise good points. It would be interesting to assess the variability in surgical/medical outcomes based on population, which could contribute to the observed treatment resistant PCG in some patients. However, the original article indicates it is the change in protein structure that is leading to these observations (which we added that information to the text (line 145- 146). Our goal is to highlight this variability and address possible explanations for its existence. While surgical and medical outcomes is an interesting area, based on the pathophysiology of PCG, it is likely that the contributing factors to a resistant PCG phenotype is cemented developmentally. This makes global genome analysis of each patient an interesting area for future studies. We added body to the discussion to address this possibility. The term mutation is used to be consistent with the primary literature. Thank you very much for taking the time to give us very helpful suggestions.
